# Impaired Neutralizing Antibody Activity against B.1.617.2 (Delta) after Anti-SARS-CoV-2 Vaccination in Patients Receiving Anti-CD20 Therapy

**DOI:** 10.3390/jcm11061739

**Published:** 2022-03-21

**Authors:** Maximilian Töllner, Claudius Speer, Louise Benning, Marie Bartenschlager, Christian Nusshag, Christian Morath, Martin Zeier, Caner Süsal, Paul Schnitzler, Wilhelm Schmitt, Raoul Bergner, Ralf Bartenschlager, Hanns-Martin Lorenz, Matthias Schaier

**Affiliations:** 1Department of Nephrology, University of Heidelberg, 69120 Heidelberg, Germany; claudius.speer@med.uni-heidelberg.de (C.S.); louise.benning@med.uni-heidelberg.de (L.B.); christian.nusshag@med.uni-heidelberg.de (C.N.); christian.morath@med.uni-heidelberg.de (C.M.); martin.zeier@med.uni-heidelberg.de (M.Z.); matthias.schaier@med.uni-heidelberg.de (M.S.); 2Molecular Medicine Partnership Unit Heidelberg, European Molecular Biology Laboratory, 69120 Heidelberg, Germany; 3Department of Infectious Diseases, Molecular Virology, University of Heidelberg, 69120 Heidelberg, Germany; marie.bartenschlager@med.uni-heideberg.de (M.B.); ralf.bartenschlager@med.uni-heidelberg.de (R.B.); 4Transplant Immunology Research Center of Excellence, Koç University Hospital, Istanbul 34010, Turkey; csusal@ku.edu.tr; 5Department of Infectious Diseases, Virology, University of Heidelberg, 69120 Heidelberg, Germany; paul.schnitzler@med.uni-heidelberg.de; 6Center for Renal Diseases, 69469 Weinheim, Germany; w.schmitt@nierenzentrum-weinheim.de; 7Clinical Center Ludwigshafen, Department of Internal Medicine A, 67036 Ludwigshafen, Germany; bergnerr@klilu.de; 8German Center for Infection Research (DZIF), Heidelberg Partner Site, 69120 Heidelberg, Germany; 9Division Virus-Associated Carcinogenesis, German Cancer Research Center (DKFZ), 69120 Heidelberg, Germany; 10Division of Rheumatology, Department of Medicine V, University of Heidelberg, 69120 Heidelberg, Germany; hanns-martin.lorenz@med.uni-heidelberg.de

**Keywords:** COVID-19, Rituximab, vaccination

## Abstract

Background: To characterize humoral response after standard anti-SARS-CoV-2 vaccination in Rituximab-treated patients and to determine the optimal time point after last Rituximab treatment for appropriate immunization. Methods: Sixty-four patients who received Rituximab within the last seven years prior to the first anti-SARS-CoV-2 vaccination were recruited in a prospective observational study. Anti-S1 IgG, SARS-CoV-2 specific neutralization, and various SARS-CoV-2 target antibodies were determined. A live virus assay was used to assess neutralizing antibody activity against B.1.617.2 (delta). In Rituximab-treated patients, CD19^+^ peripheral B-cells were quantified using flow cytometry. Results: After second vaccination, all antibodies were significantly reduced compared to healthy controls. Neutralizing antibody activity against B.1.617.2 (delta) was detectable with a median (IQR) ID_50_ of 0 (0–1:20) compared to 1:320 (1:160–1:320) in healthy controls (for all *p* < 0.001). Longer time period since last Rituximab administration correlated with higher anti-SARS-CoV-2 antibody levels and a stronger neutralization of B.1.617.2 (delta). With one exception, only patients with a CD19^+^ cell proportion ≥ 1% had detectable neutralizing antibodies. Conclusion: Our data indicate that a reconstitution of the B-cell population to >1% seems crucial in developing neutralizing antibodies against SARS-CoV-2. We suggest that anti-SARS-CoV-2 vaccination should be administered at least 8–12 months after the last Rituximab treatment for sufficient humoral responses.

## 1. Introduction

Lower humoral immune responses in patients receiving immunosuppressive maintenance therapy have already been shown in influenza and hepatitis B vaccinated individuals and were confirmed soon after vaccination started for anti-SARS-CoV-2 vaccinees [1,2,3].

Especially patients with anti-CD20 maintenance therapy, namely Rituximab (RTX), seem to develop only minimal humoral responses after anti-SARS-CoV-2 vaccination and are therefore at particular risk for severe COVID-19 breakthrough infection [4,5]. Anti-CD20 treatment is effective against various systemic diseases and minimizes CD20-positive B-lymphocytes without depleting bone marrow or mature, long-lived plasma cells. This mode of action, however, leads to a strong reduction in humoral immune responses after exposure to new virus or vaccine antigens. As increased immune responses have already been reported in patients receiving influenza vaccination after delayed treatment, establishing an appropriate time interval between the last administration of anti-CD20 and anti-SARS-CoV-2 vaccination may be useful to improve vaccine response in these patients [6]. Furthermore, the persistence of circulating anti-SARS-CoV-2 antibodies after vaccination and subsequent first anti-CD20 treatment is unknown. The latter, however, is crucial in order to develop strategies for booster vaccination.

Given that mutations in the glycosylated SARS-CoV-2 spike protein lead to partial immune escape of new variants of concern (VoCs) such as the B.1.617.2 (delta) variant, higher antibody levels are required for adequate virus neutralization [7]. The major target of neutralizing antibodies that block SARS-CoV-2 entry into host cells via the angiotensin-converting enzyme 2 (ACE2) host cell receptor represents the receptor-binding domain protein (RBD) [8].

Previous studies describe the humoral and cellular immune response and, more recently, variant neutralization in patients receiving Rituximab maintenance therapy [9,10,11].

In this study, we investigated in-depth humoral responses after standard SARS-CoV-2 vaccination in 64 patients with Rituximab treatment. We aimed to determine the optimal time point for anti-SARS-CoV-2 vaccination after the last anti-CD20 depletion therapy to improve on immunization protocols against VoCs in this vulnerable cohort.

## 2. Materials and Methods

### 2.1. Study Design

This prospective, observational cohort study was conducted at four German autoimmune centers. It includes 64 patients with systemic rheumatic diseases who received Rituximab therapy within the last seven years before their first anti-SARS-CoV-2 vaccination.

From February to September 2021, a total of 91 serum samples from 64 patients were collected after a median (IQR) of 24 (20–36) days after first and a median (IQR) of 24 (21–44) days after second vaccination to determine humoral vaccine response in 27 and 64 participants, respectively (Figure 1).

Anti-spike S1 IgG and SARS-CoV-2 specific surrogate neutralizing antibodies were assessed at both time points in all patients (Figure 1). A bead-based assay to analyze different antibodies against different SARS-CoV-2 target epitopes and common cold coronaviruses target epitopes was performed in all patients at the second time point. Patients were defined as triple positive, if all three commercial markers for humoral response, namely anti-spike S1 IgG, SARS-CoV-2 specific surrogate neutralizing antibodies and anti-RBD antibodies were above the predefined cut-off values. Non-responder, single and double positives were classified the same way. Neutralizing antibody activity against the variant of concern B.1.617.2 (delta) was determined using a live virus neutralization assay in all 21 patients with an anti-S1-IgG above the cut-off value, in 20 patients with insufficient anti-S1 response and in all 36 healthy controls. We detected antibodies or antibody activity against the SARS-Cov-2 wild type with commercially available antibody tests. Antibody responses against the B.1.617.2 (delta) variant were quantified with a live virus assay.

After the first and second vaccination, adverse events were assessed with a standardized questionnaire inquiring about local and systemic adverse reactions and use of anti-inflammatory medication. The questionnaire is presented in the Appendix A. Kidney function and disease activity, if possible, were determined serologically at both time points. CD4^+^ and CD19^+^ lymphocyte subpopulations were determined once after second vaccination in 50 patients. Lymphocytes were not assessed in healthy controls.

Age and vaccine-matched healthy individuals served as controls for characterization of humoral responses and adverse reactions after the first (*n* = 21) and after the second vaccination (*n* = 36). Healthy individuals were selected by frequency matching. In *n* = 15 healthy controls, samples could only be collected after second vaccination. Patients and controls with a history of previous SARS-CoV-2 infection or positivity for anti-nucleocapsid antibodies were excluded.

The study was approved by the Ethics Committee of the University of Heidelberg: S-416/2021 and conducted in accordance with the Declaration of Helsinki. Participants gave informed consent to participate in the study before taking part.

### 2.2. IgG Antibodies against SARS-CoV-2 Spike S1 and Nucleocapsid

To measure the IgG response against the S1 protein, we used the SARS-CoV-2 IgG Assay (Siemens, Eschborn, Germany). A semi-quantitative index of <1 was classified as negative, and a value of ≥1 as positive. This cut-off for detection gives a specificity of 100% with a sensitivity of 98% for detecting anti-S1 IgG antibodies. IgG response against the nucleocapsid protein was determined by the Elecsys anti-SARS-CoV-2 assay (Roche, Mannheim, Germany). Both assays were performed according to the manufacturer’s protocol.

### 2.3. SARS-CoV-2 Specific Surrogate Neutralizing Antibodies

Surrogate Neutralizing antibodies were determined with a surrogate neutralization test (Medac, Wedel, Germany), as we described previously [12,13,14,15]. This test simulates the virus-host cell interaction by direct protein-protein interaction using purified, viral RBD protein and angiotensin-converting enzyme 2 (ACE2) from the host cell [16]. Optical density was measured in each well, and the percent inhibition (%) was calculated as follows:$$\mathrm{Inhibition}=\left(1-\left(\frac{\mathrm{OD}\;\mathrm{value}\;\mathrm{of}\;\mathrm{Sample}}{\mathrm{OD}\;\mathrm{value}\;\mathrm{of}\;\mathrm{Negative}\;\mathrm{Control}}\right)\right)\times 100\%\;$$

The test achieves 99.9% specificity with 95–100% sensitivity to detect surrogate neutralizing antibodies with a cut-off of ≥30% inhibition of RBD:ACE2 binding defining positivity.

### 2.4. IgG Antibodies against Different SARS-CoV-2 and Common Cold Coronaviruses Target Epitopes

We used a multiplex bead-based assay for the Luminex platform (LabScreen COVID Plus, One Lambda Inc., West Hill, CA, USA) to differentiate IgG antibodies against different SARS-CoV-2 target epitopes [17]. Antibodies against the nucleocapsid protein and 4 different fragments of the SARS-CoV-2 spike protein, namely the full spike protein, the spike S1, the spike S2, and the RBD of the spike protein, were detected. Furthermore, antibody reactivity was measured against the S1 fragments of five other coronaviruses, namely HCoV-229E, HCoV-HKU1, HCoV-NL63, HCoV-OC43, and SARS-CoV-1. The mean fluorescence activity (MFI) was analyzed on a Luminex 200 device (Luminex Corporation, Noord-Brabant, The Netherlands). Cut-off values are shown for each of the 10 proteins in Appendix A Appendix A.

### 2.5. Neutralization against the B.1.617.2 (Delta) Variant of Concern

Titration experiments on VeroE6 cells were used to measure neutralization titers against B.1.617.2 (delta), as previously described by us [13,18]. Nasopharyngeal and oropharyngeal swabs of PCR-confirmed B.1.617.2 (delta) positive patients served as sources for virus isolation using VeroE6 cells; virus stocks were generated in the same cell line and titers were determined as described previously by us and others [13,19]. Two-fold serial dilutions of patient sera were mixed with the virus (10e4 PFU/mL). After 1 h incubation at 37 °C, the mixture was added to VeroE6 cells for 24 h. Medium was removed, cells were fixed in the plates with 5% formaldehyde, and virus replication was determined by an in-cell ELISA detecting viral nucleoprotein via immunostaining. Cells infected in the absence of patient serum (100% infection) and non-infected cells (0% infection) served as controls to normalize values and set the assay background, respectively. The ID_50_ refers to the serum dilution that reduces infection of cells by 50% with the cut-off for detection being the 1:10 dilution of a given serum.

### 2.6. Statistics

Data are expressed as median and interquartile range (IQR) or number (N) and percent (%). Continuous variables were compared using the Mann-Whitney U test for the analysis of two groups. For the analysis of three or more groups, Kruskal-Wallis test with Dunn’s post-test was applied. Spearman’s rho as a nonparametric measure of rank correlation was calculated to describe the relationship between two different tests detecting humoral immunity. Statistical significance was assumed at a *p*-value < 0.05. The statistical analyses were performed using GraphPad Prism version 9.3.0 (GraphPad Software, San Diego, CA, USA).

## 3. Results

### 3.1. Study Population

We compared 64 patients who received Rituximab within the last seven years with 36 healthy controls. Patients and healthy controls were age-matched with a median (IQR) age of 58 (50–69) years in the patient group compared to a median (IQR) age of 59 (45–62) years in the control group (*p* = 0.4). While 54 patients received two mRNA vaccine doses (47 BNT162b2 and seven mRNA-1273), four were heterologously and six homologously vaccinated with ChAdOx1 nCoV-19. All healthy controls were homologously vaccinated, 33 with BNT162b2 and three with ChAdOx1 nCoV-19. Humoral responses after first vaccination were assessed after a median (IQR) of 24 (20–36) days in patients compared to 18 (17–21) days in healthy controls. Second vaccination was applied after a median (IQR) of 35 (28–42) days in patients compared to 21 (21–21) in healthy controls. Humoral responses were assessed a median (IQR) of 24 (21–44) days after second vaccination in patients compared to 21 (19–22) days in healthy controls. The German authorities changed vaccine recommendations to a more extended period between both vaccinations leading to the difference in vaccine interval between patients and controls.

Patients received the last Rituximab treatment with a median (IQR) of 5 (4–10) months before their first vaccine dose. A total of 88% took additional immunosuppressive drugs as maintenance therapy. The main difference between responders and non-responders was the time since last rituximab administration with a median (IQR) of 21 (9–31) months in responders compared to a median (IQR) of 5 (4–7) months in non-responders. Detailed baseline characteristics of patients and controls are shown in Table 1.

We observed significantly more adverse reactions after the second vaccine dose compared to the first vaccine dose in both groups (*p* = 0.002). In addition, healthy controls reported significantly more adverse reactions than individuals with immunosuppressive medication (*p* = 0.004). Detailed adverse reactions in both groups are shown in Appendix A Appendix A.

### 3.2. Anti-S1 IgG and Surrogate Neutralizing Antibodies after First and Second Anti-SARS-CoV-2 Vaccination

After the first vaccination, patients had significantly lower anti-S1 IgG and surrogate neutralizing antibodies with a median (IQR) anti-S1 index of 0.1 (0.1–0.19) compared to 8 (2.6–13.8) in healthy controls and a median (IQR) inhibition of 22% (12–24) compared to 46% (43–73), respectively (for all *p* < 0.001) (Figure 2A,B).

After the second vaccination, anti-S1 IgG and surrogate neutralizing antibodies increased significantly in both groups (for all *p* < 0.001). However, only 21/64 (33%) patients had detectable anti-S1 IgG above the predefined threshold with a median (IQR) of 0.1 (0.1–2.3) compared to 36/36 (100%) healthy controls with a median (IQR) of 150 (74.6–278.1) (*p* < 0.001). Inhibition of surrogate neutralizing antibodies was above the cut-off in 33/64 (52%) patients with a median (IQR) inhibition of 30% (20–48) compared to 36/36 (100%) healthy controls with a median (IQR) of 98% (97–98) (*p* < 0.001) (Figure 2C–F).

### 3.3. IgG Antibodies against Different SARS-CoV-2 Epitopes and Other Common Cold Coronaviruses after the Second Vaccination

In the present study we further determined the IgG antibody response against different SARS-CoV-2 target epitopes after the second anti-SARS-CoV-2 vaccination in a bead-based multiplex assay. As shown in Figure 3A, patients with a history of Rituximab treatment showed significantly lower MFI values against all SARS-CoV-2-specific proteins. No healthy control or patient had an MFI value above the cut-off for anti-nucleocapsid antibodies indicating no previous SARS-CoV-2 infection.

Antibody reactivity against three different spike proteins was significantly lower in patients compared to healthy controls with a median (IQR) MFI of 2995 (0–17,700) compared to 23,152 (22,735–23,757) for full spike, 0 (0–5463) to 15,659 (14,103–17,300) for spike S1, and 189 (0–3520) to 7866 (5604–12,905) for spike S2, respectively (for all *p* < 0.001). Anti-RBD reactivity was also significantly lower in patients with a median (IQR) MFI of 195 (0–9226) compared to 19,910 (18,110–21,051) in healthy controls (*p* < 0.001) (Figure 3A). Cumulative humoral response against different SARS-CoV-2 specific targeted epitopes was significantly lower in patients with a median (IQR) MFI of 4792 (350–33,174) compared to 66,645 (60,837–73,915) in healthy individuals (*p* < 0.001) (Figure 3B).

Furthermore, we determined IgG antibody response against different common cold coronaviruses, namely HCoV-229E, HCoV-HKU1, HCoV-NL63, HCoV-OC43, and SARS-CoV Spike S1. The results revealed no significant difference in antibody reactivity, in patients compared to the healthy controls, against HCoV-229E, HCoV-HKU1, HCoV-NL63, HCoV-OC43 with a median (IQR) MFI of 9022 (5779–11,664) compared to 9502 (7878–11,470), 4870 (3298–6497) to 4835 (3091–5932), 4411 (2984–5602) to 4223 (2223–5033), and 3593 (2129–5818) to 4,006 (2511–5108). With a median (IQR) of 0 (0–88) compared to a median (IQR) 1274 (891–1871) in healthy controls, SARS-CoV Spike 1 IgG antibodies in patients were significantly lower (*p* < 0.001) (Figure 3C).

### 3.4. Neutralizing Antibody Activity against B.1.617.2 (Delta) after Second Vaccination and Correlation with Commercial Assays and CD19^+^ Cell Proportion

The ID_50_ values (equates the serum dilution that reduces infection of cells by 50%) against the B.1.617.2 (delta) variant were significantly lower in patients with a median (IQR) ID_50_ of 0 (0–1:20) compared to 1:320 (1:160–1:320) in healthy controls (*p* < 0.001) (Figure 4A). While 36/36 (100%) of healthy individuals showed neutralizing activity against B.1.617.2 (delta), 0/20 (0%) of all patients without and 13/21 (62%) patients with detectable anti-S1 IgG above the cut-off value were able to neutralize B.1.617.2 (delta) at least at the minimal dilution. Patients who did not have detectable antibodies above the predefined levels in all three tests (anti-S1 IgG, surrogate neutralizing, and anti-RBD antibodies) did not show any neutralizing antibody activity. In 13/17 (76%) triple-positive patients, neutralization of B.1.617.2 (delta) was detected with a median ID_50_ (IQR) of 1:20 (1:10–1:160) (Figure 4B). The distribution of non, single, double, and triple positives in the patient group is shown in Appendix A Appendix A.

The ID_50_ values obtained with patient sera correlated with all three determined commercial antibody tests (anti-S1-IgG, surrogate neutralizing antibodies, and anti-RBD antibodies) with a correlation coefficient of 0.75, 0.84, and 0.87, respectively (for all *p* < 0.001) (Figure 4C–E). Only values above the cut-offs were used for correlation.

Furthermore, CD19^+^ proportion correlated with neutralization of B.1.617.2 (delta) with a correlation coefficient of 0.68 (*p* < 0.001) (Figure 4F).

### 3.5. Humoral Immune Response Depending on the Last Rituximab Treatment Prior to the First Anti-SARS-CoV-2 Vaccination

Stronger humoral responses correlated with a longer time since last Rituximab treatment. All three assessed commercially available markers of humoral response, i.e., anti-S1 IgG, surrogate neutralizing and anti-RBD antibodies, increased over time with a correlation coefficient of 0.63, 0.45 and 0.56, respectively (for all *p* < 0.001) (Figure 5A–C). Furthermore, the cumulative antibody response against different SARS-CoV-2 specific epitopes and the proportion of CD19^+^ cells increased over time with a correlation coefficient of 0.57 and 0.30, respectively (for all *p* < 0.001) (Figure 5D,E).

Neutralizing antibody activity against the B.1.617.2 (delta) variant was first detectable in a patient eight months after last Rituximab treatment, and between 8–12 months after last Rituximab treatment, 5/11 (45%) patients had neutralizing antibody activity against the B.1.617.2 (delta) variant. With an increasing number of patients showing anti-S1 IgG antibodies above the cut-off, 8/11 (72%) patients 12 months and 5/6 (83%) patients 24 months after the last Rituximab treatment showed neutralizing activity (Figure 5C). Finally, after 19 months, 8/8 (100%) patients with an IgG-Index above the predefined cut-off had detectable neutralizing antibody activity (Figure 5F).

The median (IQR) proportion of CD4^+^ cells to total lymphocytes was 57% (38–71). The proportion of CD19^+^ cells was predictably low, with a median (IQR) of 0% (0–1). Whereas the proportion of CD4^+^ cells did not correlate with humoral response, CD19^+^ cells were associated with levels of anti-S1-IgG, surrogate neutralizing, and anti-RBD antibodies with correlation coefficients of 0.46, 0.50, and 0.53, respectively (for all *p* < 0.001) (Appendix A Appendix A).

No disease-specific effects on the humoral response were found in correlation with time since last Rituximab administration. However, CD19^+^ cell recovery was significantly delayed in patients with small vessel vasculitis compared to patients with connective tissue disease, rheumatoid arthritis or other diseases (*p* = 0.02) (Appendix A Appendix A). Furthermore, antimetabolites were associated with reduced humoral response and B-cell recovery compared to Methotrexat, corticosteroids and other immunosuppressive co-medications (Appendix A Appendix A).

## 4. Discussion

Patients with a history of Rituximab therapy are at particular risk for severe SARS-CoV-2 infection [20]. Determining the time point after Rituximab administration at which anti-SARS-CoV-2 vaccination leads to neutralizing antibody activity against variants of concern is urgently needed for treatment-adapted immunization protocols in this vulnerable group.

This is one of the first studies to evaluate VoC-adapted humoral responses using a live virus neutralization assay in patients with Rituximab therapy. Our data show impaired humoral responses in Rituximab-treated patients compared to healthy controls. Neutralizing antibody activity against B.1.617.2 (delta) was not detected in patients if the last Rituximab treatment was administered less than seven months before the first vaccination. Moreover, the intensity of a vaccine-induced humoral response correlated with an increasing time interval after last Rituximab administration.

Impaired humoral response after vaccination in patients receiving Rituximab has already been shown in pre-pandemic studies [1,6,21,22]. Recent data confirmed a significantly reduced humoral response after standard SARS-CoV-2 vaccination in patients with systemic diseases receiving Rituximab [4,10,11,23]. However, drug-specific differences in humoral responses remain difficult to determine, especially when different immunosuppressive medications are combined. Rasselt et al. showed that Rituximab administration was particularly frequently associated with a lack of seroconversion (90%), whereas Mycophenolate-mofetil (25%) had a lesser impact. However, additional negative effects of immunosuppressive co-medication should not be neglected [24]. B-cell repopulation and time between last Rituximab treatment prior to vaccination seems to be crucial as König et al. and others demonstrated [25,26]. The average time to B-cell repopulation ≥ 1% after Rituximab treatment is 6–12 months [27,28], and a peripheral CD19^+^ B-cell population of ≥1% seems to be necessary to generate detectable IgG response following vaccination [23,25,26,29,30]. For this reason, delaying the subsequent anti-CD20 treatment may lead to improved humoral response without risking increased disease breakthrough activity [31]. Similar to previous investigations, we observed disease-specific significant differences in B-cell recovery between patients with rheumatoid arthritis and patients with small vessel vasculitis [32]. Therefore, disease-specific delayed seroconversion in patients with small vessel vasculitis cannot be excluded.

The most common tests are commercial kits detecting response to the SARS-CoV-2 wild type. However, the vaccine developed against the wild type already showed a reduced neutralization against VOC B.1.617.2 (delta) in healthy individuals [7]. Hence, positive rates in commercial tests are expected to match lower activity of neutralizing antibodies against VOCs like B.1.617.2 (delta). Recently, Hadjadj et al. investigated neutralization of B.1.1.7 (alpha) and B.1.617.2 (delta) in immunocompromised patients and did not detect neutralizing activity in Rituximab-treated patients. This finding is consistent with our data, as the last Rituximab administration with a median IQR of 13.5 (0–117.5) days lies within the six-month interval mentioned above [11]. So far, there have been no other reports of live virus neutralization in patients with discontinued or suspended anti-CD20 therapies.

The determined IgG antibodies against common cold coronaviruses revealed no significant difference between healthy controls and patients treated with Rituximab. Therefore, the adequate humoral response observed may be attributed to long-lived plasma cells that are not depleted by Rituximab treatment. This finding suggests that patients who received anti-CD20 therapy for the first time after being fully vaccinated against SARS-CoV-2 may have a sustained antibody response. Since only healthy controls had antibodies against SARS-CoV-1, current SARS-CoV-2 vaccines are likely to result in cross-reactive immune response against SARS-CoV-1 [33].

Our data show an impaired humoral response in patients who had received Rituximab within six months before vaccination. However, only 26 patients who had received their last treatment > 6 months prior to the first vaccination were recruited. Further studies including more patients in this intriguing period after Rituximab treatment are needed to confirm our results. As most patients received immunosuppressive co-medication, the effect on immune response cannot be solely attributed to Rituximab. Moreover, no significant difference between different vaccine schemes was detected, since immune response was deficient and larger patient groups would have been required. Nevertheless, homologous ChAdOx1 nCoV-19 vaccination led to inferior immune response in healthy controls, while heterologous vaccination was found to be equal to homologous mRNA vaccination [12]. Furthermore, it has already been shown that a longer interval between prime and booster vaccination leads to a better immune response. In this regard, the difference in humoral response due to the shorter vaccination intervals in the control group could be even more significant than demonstrated in this study [34].

While cellular responses in B-cell depleted patients seemed to be only slightly reduced, several studies revealed heterogeneous results. Mrak et al. and others demonstrated a 58–83% T-cell response using an ELISPOT assay [9,10,11,29], whereas a 20% cellular response was determined by Moor et al. via Interferon-gamma release assay [35]. Collectively, vaccination induces a cellular response, but further extensive studies with calibrated quantification methods need to follow. The same applies to vaccine response after third vaccination, where current data indicate a reduced humoral response even after booster vaccination in patients with recent Rituximab treatment [36,37,38].

Our results show impaired humoral response after standard anti-SARS-CoV-2 vaccination in Rituximab-treated patients. Seroconversion in commercially available assays correlates but does not equal live virus neutralization against variants of concern such as B.1.617.2 (delta). Delaying Rituximab therapy in patients with stable disease to allow restoring of the B-cell population may be a promising attempt to achieve better antibody response in the future. In addition, the proposed treatment-adapted vaccination schedules for the upcoming modified vaccine against new variants of concern such as B.1.1.529 (omicron) should be considered.

## Figures and Tables

**Figure 1 jcm-11-01739-f001:**
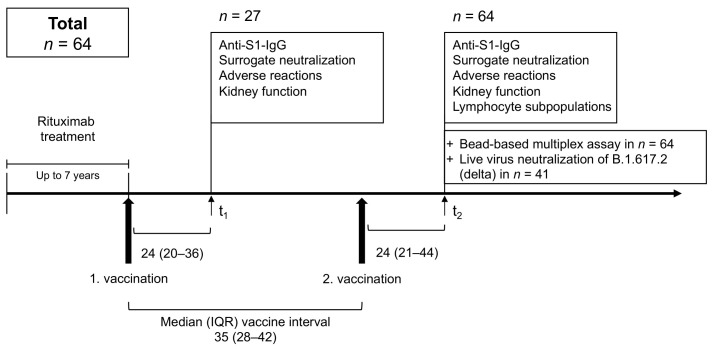
Study design to determine humoral immune responses to standard anti-SARS-CoV-2 vaccination in Rituximab-treated patients in a prospective, observational cohort study.

**Figure 2 jcm-11-01739-f002:**
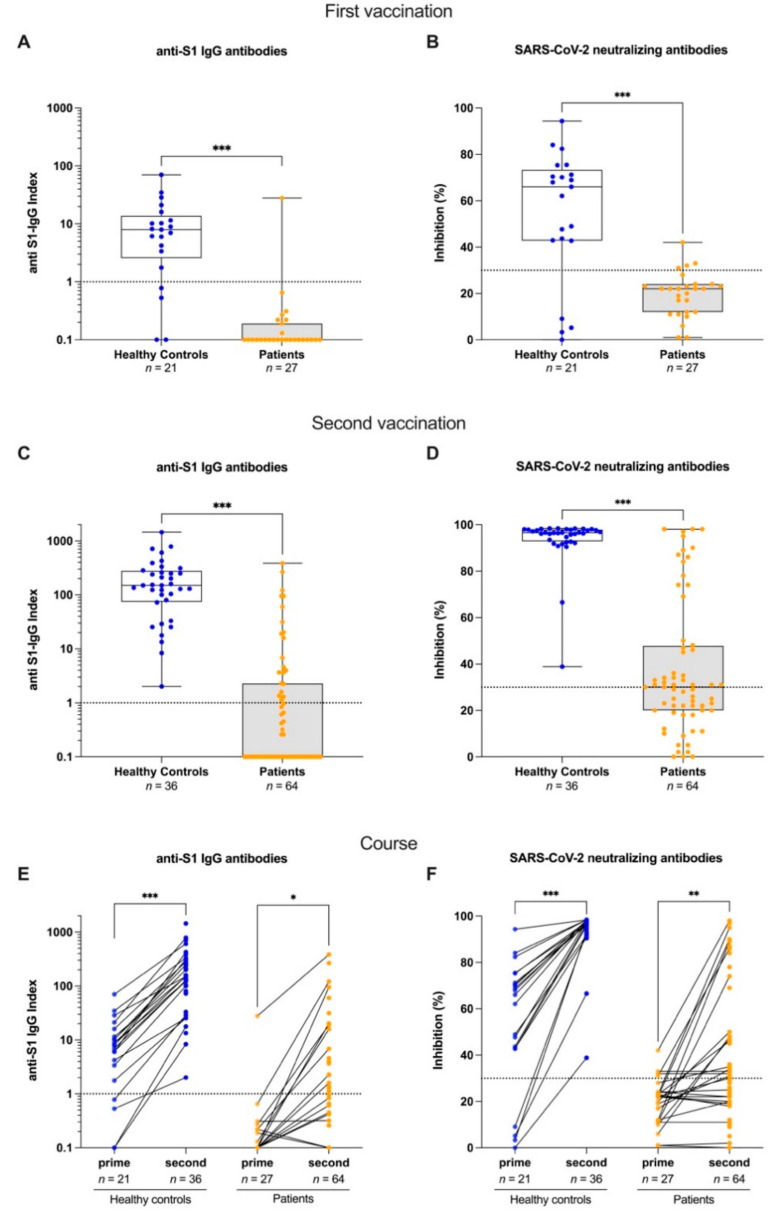
Anti-S1 IgG and surrogate neutralizing antibodies after first and second anti-SARS-CoV-2 vaccination and individual course in Rituximab-treated patients and healthy controls. (**A**) SARS-CoV-2 IgG antibodies were determined by a chemiluminescent immunoassay. Rituximab (RTX) treated patients are compared to healthy controls after first vaccination. The y-axis shows the anti-S1 IgG index, represented logarithmically. The dashed red line indicates the cut-off for detection. A semiquantitative index ≥1 was classified as positive. (**B**) Surrogate neutralizing antibodies as determined by a surrogate virus neutralization test after first anti-SARS-CoV-2 vaccination. RTX-treated patients are compared to healthy controls after first vaccination. The y-axis shows the percent (%) binding inhibition. The dashed red line indicates the cut-off for detection with a cut-off of ≥30% defining positivity. (**C**) Antibodies against SARS-CoV-2 IgG after second vaccination in RTX-treated patients and healthy controls. (**D**) Surrogate neutralizing antibodies after second vaccination in RTX-treated patients and healthy controls. (**E**) Individual course of SARS-CoV-2 IgG antibodies after first and second vaccination in healthy controls and RTX-treated patients. (**F**) Individual course of surrogate neutralizing antibodies after first and second vaccination in healthy controls and RTX-treated patients. *n*, number; *** *p* < 0.001; ** *p* < 0.01; * *p* < 0.05.

**Figure 3 jcm-11-01739-f003:**
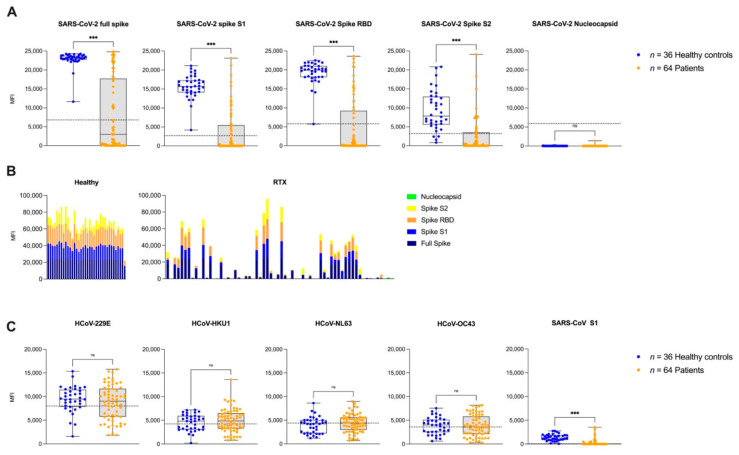
IgG antibodies against different SARS-CoV-2 and other common cold coronaviruses target epitopes after the second vaccination. (**A**) Antibody reactivity against different SARS-CoV-2 target epitopes, namely the SARS-CoV-2 full spike, spike S1, spike RBD, spike S2 protein, and the nucleocapsid protein of SARS-CoV-2, after second vaccination, as determined by a bead-based multiplex assay. The dashed red line indicates the cut-off for detection for each respective target. (**B**) Reactivity patterns for different SARS-CoV-2 target antigens. The height of each histogram bar indicates the cumulative mean fluorescence intensity (MFI) value for detected antibodies against the full spike, the S1 spike, the RBD of the spike, the S2 spike, and the nucleocapsid protein of SARS-CoV-2. (**C**) Antibody reactivity against different target epitopes of common cold coronaviruses, namely HCoV-229E, HCoV-HKU1, HCoV-NL63, HCoV-OC43, and SARS-CoV Spike S1. Cut-offs are given in the Appendix A. MFI, mean fluorescence intensity; *n*, number; RBD, receptor-binding domain; ns, not significant; *** *p* < 0.001.

**Figure 4 jcm-11-01739-f004:**
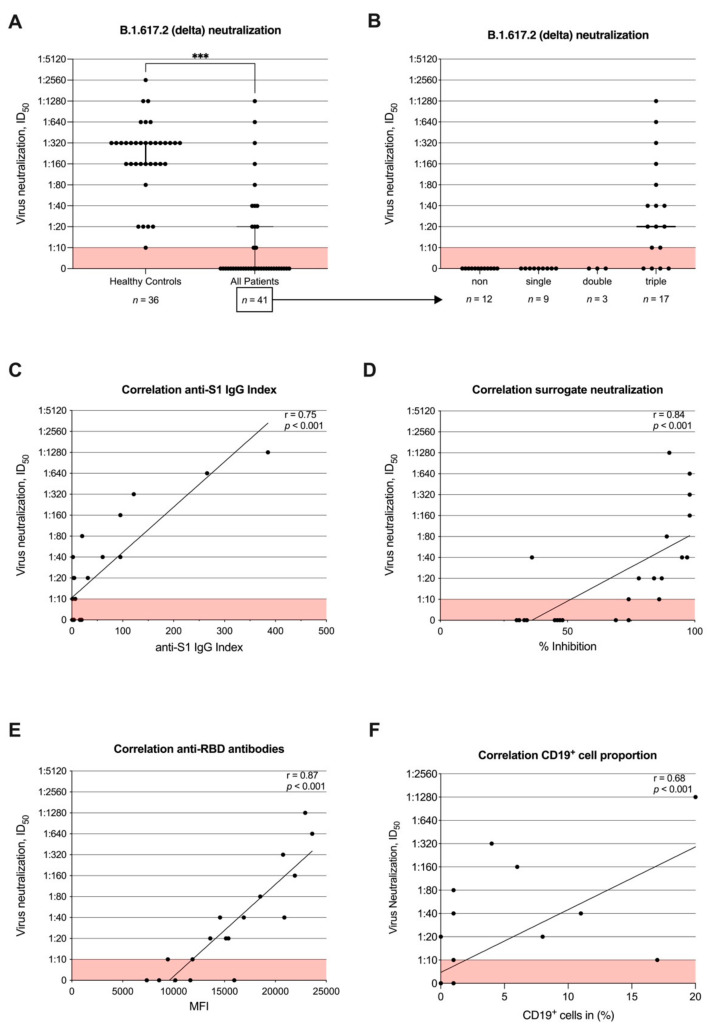
Neutralizing antibody activity against B.1.617.2 (delta) after second vaccination and correlation with commercial assays and CD19^+^ cell proportion. Titers of neutralizing antibodies against B.1.671.2 (delta) were determined in a SARS-CoV-2 infection assay using VeroE6 target cells and serial two-fold serum dilutions. (**A**) Titers of neutralizing antibodies against B.1.671.2 (delta) in RTX-treated patients and healthy controls. (**B**) Titers of neutralizing antibodies against B.1.671.2 (delta) in RTX-treated patients separated according to seroconversion in anti-S1 IgG, SARS-CoV-2-specific neutralizing antibodies, and anti-RBD antibodies. (**C**–**F**) Correlation of titers of neutralizing antibodies against B.1.671.2 (delta) with anti-S1 IgG, SARS-CoV-2-specific neutralizing antibodies, anti-RBD antibodies, and CD19^+^ cell proportion of total lymphocytes evaluated by Spearman correlation analysis, respectively. MFI, mean fluorescence intensity; *n*, number; RBD, receptor-binding domain; *** *p* < 0.001.

**Figure 5 jcm-11-01739-f005:**
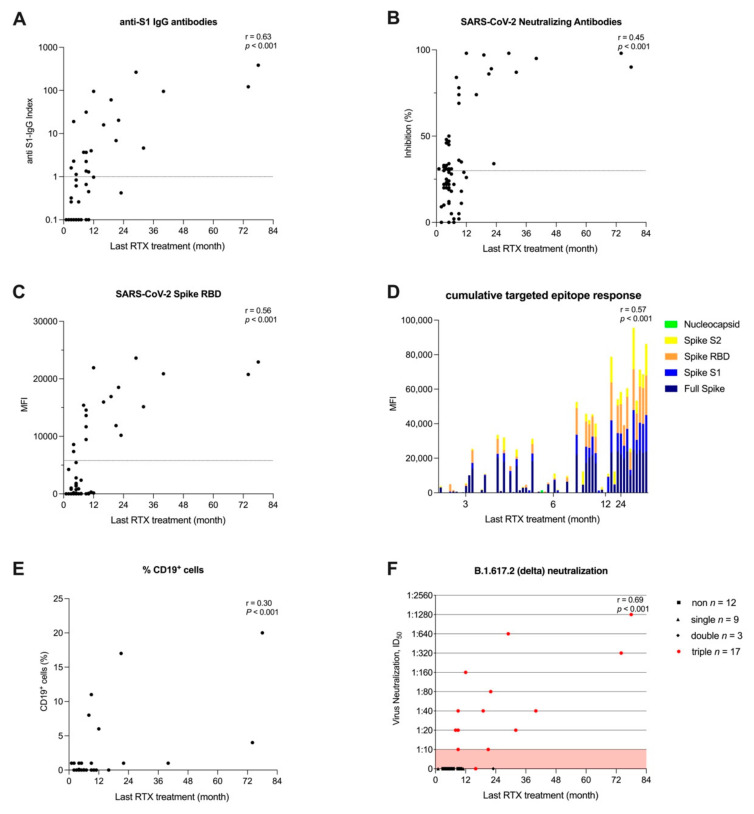
Humoral response depending on the last RTX treatment prior to the first anti-SARS-CoV-2 vaccination. (**A**) Anti-S1 IgG antibodies depending on the last RTX administration. The x-axis displays the time between the last RTX administration and the first vaccination. The y-axis shows the anti-S1 IgG index, represented logarithmically. (**B**) Surrogate neutralizing antibodies depending on the last RTX administration. (**C**) Anti-RBD antibodies depending on the last RTX administration. (**D**) Reactivity patterns for different SARS-CoV-2 target antigens depending on the last RTX administration. (**E**) CD19^+^ proportion of total lymphocytes depending on the last RTX administration. (**F**) Titers of neutralizing antibodies against B.1.671.2 (delta) according to seroconversion in anti-S1 IgG, SARS-CoV-2-specific neutralizing antibodies, and anti-RBD antibodies depending on the last RTX administration. MFI, mean fluorescence intensity; *n*, number; RBD, receptor-binding domain.

**Table 1 jcm-11-01739-t001:** Baseline characteristics and vaccination history of patients and healthy controls. Patients were additionally divided into responders and non-responders. Response was defined as detectable neutralizing antibody activity against the B.1.617.2 (delta) variant.

	Healthy Controls	Patients	Responders	Non-Responders
	(*n* = 36)	(*n* = 64)	(*n* = 13)	(*n* = 51)
Sex				
Female	24 (67%)	39 (59%)	9 (69%)	30 (59%)
Male	12 (33%)	25 (41%)	4 (31%)	21 (41%)
Median age, years	59 (45–62)	58 (50–69)	57 (50.5–63.5)	62 (51–70)
Vaccine regimen				
Homologous BNT162b2	33 (92%)	47 (74%)	6 (46%)	41 (80%)
Homologous mRNA-1273	0 (0%)	7 (11%)	6 (46%)	1 (2%)
Heterologous ChAdOx1 nCoV-19/BNT162b2	0 (0%)	3 (5%)	0 (0%)	3 (6%)
Heterologous ChAdOx1 nCoV-19/mRNA-1273	0 (0%)	1 (1%)	0 (0%)	1 (2%)
Homologous ChAdOx1 nCoV-19	3 (8%)	6 (9%)	1 (8%)	5 (10%)
Median (IQR) time since first vaccination, days	18 (17–21)	24 (20–36)	24 (21–32)	25 (19–37)
Median (IQR) time since second vaccination, days	21 (19–22)	24 (21–44)	29 (22–41)	23 (21–47)
Median (IQR) time between both vaccinations, days	21 (21–21)	35 (28–42)	42 (35–42)	35 (26–42)
Median (IQR) time since last RTX therapy, months	-	5 (4–10)	21 (9–36)	5 (4–7)
Immunosuppressive co-medication		*n* = 56 (88%)	*n* = 13 (100%)	*n* = 43 (84%)
Glucocorticoids	-	35/56 (63%)	5/13 (38%)	30/43 (70%)
Antimetabolites	-	30/56 (54%)	6/13 (46%)	24/43 (56%)
Antimalarials	-	5/56 (9%)	3/13 (23%)	2/43 (5%)
Methothrexate	-	10/56 (18%)	3/13 (23%)	7/43 (16%)
Biologicals	-	1/56 (2%)	0/13 (0%)	1/43 (2%)
Other immunosuppression	-	2/56 (4%)	0/13 (0%)	2/43 (5%)
Disease				
ANCA-associated vasculitis	-	35 (55%)	8 (61%)	27 (52%)
Systemic lupus erythematosus	-	7 (11%)	3 (23%)	4 (8%)
Rheumatoid arthritis	-	5 (8%)		5 (10%)
Cryoglobulinemic vasculitis	-	5 (8%)		5 (10%)
Myositis	-	4 (6%)	1 (8%)	3 (6%)
IgG4-associated disease	-	3 (5%)		3 (6%)
Sjögren’s syndrome	-	2 (3%)	1 (8%)	1 (2%)
Membranous nephropathy	-	1 (1%)		1 (2%)
Systemic sclerosis	-	1 (1%)		1 (2%)
Thrombotic thrombocytopenic purpura	-	1 (1%)		1 (2%)

## Data Availability

The data of this study are available on request from the corresponding authors.

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
