# Peer review of "Impaired Neutralizing Antibody Activity against B.1.617.2 (Delta) after Anti-SARS-CoV-2 Vaccination in Patients Receiving Anti-CD20 Therapy"

_jcm, 2022, doi:10.3390/jcm11061739_

Round 1

Reviewer 1 Report

In this manuscript, authors present the antibody responses induced by COVID-19 vaccine in patients under Rituximab therapy. This manuscript lacks clarity on its objective, findings and novelty. Suggestions to improve the manuscript are below:

  1. The objective of the study is not clear. In the introduction, authors need to highlight the several studies related to Rituximab treatment and COVID vaccination already performed, explain how this study is novel and what research question (gap) it answers. Studies have already shown that patients under Rituximab treatment will have reduced antibody responses, including against delta variant.    
  2.  The study design and methodology is also not clear. Why are there different numbers of healthy cohorts after the first and second vaccination? The antibody responses just mention SARS-CoV-2. Are those antigens relevant to the ancestral (vaccine) virus, other than where specifically mentioned as delta? 
  3. Results need to be represented focussing more on the objective of the study. Though the CD4+ and CD19+ cells are mentioned their population comparison is not shown for healthy and test cohorts. It is also not clear whether they were determined before vaccination. How did the cell population and antibody responses varied as per the time of last Rituximab treatment? What is the relevance of data related to the seasonal coronaviruses? The patients are on other immunosuppressive medication and their effect needs to be explored. Likewise, the treatment was for different diseases. Was there any differential outcome according to the disease? The adverse events appear to be more in healthy cohorts, combine both figures together to make it look clearer. The supplementary figure S3 should be described only after the main figures of antibody responses, and B cells are described. 

Author Response

Please see the attachemnet. Line references refer to the proof manuscript.

Reviewer 2 Report

This a well performed and well described study.

Few comments:

  • The patient population is rather heterogenous with differences in autoimmune conditions and concomitant immunosupressive medications. Without a control group of non-rituximab exposed autoimmune patients it is difficult to attribute all of the impaired humoral responses seen in this study to previous rituximab treatment. However, this remains a clinically relevant population.
  • The immunological assays are appropriate including the use of live neutralisation assay (against delta)
  • The differences in vaccine interval between the patient and healthy control group should be expanded. Differences in boosting intervals are well known to impact on the humoral response of these vaccines - which in this case could mean the autoimmune group actually perform even worse than a healthy volunteer group with the same vaccine interval.
  • It would be good to add in a breakdown of the baseline characterstics of the post-vaccination antibody responders vs non-responders to the paper
  • Looks like reduced reactogenicity in rituximab group which is an interesting finding - although probably due to co-administration of steroids / other immunosupressants in the rtx group vs healthy controls. 
  • Abstract is good but can you add in that it's a prospective observational study
  • existing baseline characteristics table is a bit confusing in its description of vaccinations - (i.e. why are there only 27 first vaccinations if there are 64 patients with 2 doses of vaccines in the trial?). Also make sure percentage rounding is consistent in this table (i.e. why is n=1 given as 2% for biologicals but n=1 given as 1% for systemic sclerosis etc)

very minor additional points:

  • Fix "adverce" reaction typo in figure 1

Author Response

Please see the attachment. Line references refer to the proof manuscript.

Reviewer 3 Report

This is an interesting paper that reveals impaired neutralizing antibody activity against the SARS-CoV-2 delta variant in patients that were previously treated with rituximab. Here are some comments:

  1. Figure 3. The color labels referring to A and C are obviously the ones shown at the right end. Moving them closer to A and repeating them again for C could be a good idea
  2. I think that the legend of Figure 4 seems to be separated from the legend and it is easily mistaken as a part of the results section. It should be added to the body of the Figure’s legend
  3. Please increase the letter size of r and p in the correlation analyses in figures 4 and 5, as they are very hard to see
  4. Table 1: The words ‘sex’ and ‘disease’ at the left column are displaced
  5. Please mention the time period when the study was performed
  6. Figure 2 E & F: Please mention at the X-axis which group is which. I guess the one at the right is the RTX group, but this should be mentioned
  7. Figure 5 A, B, and possibly C: Using the non-parametric Spearman correlation ρ is correct, however, the lines in the graphs may be somewhat misleading since the association here appears to be non-linear (exponential). To my knowledge, Spearman correlation does not necessarily imply linearity
  8. Did you notice any difference in B-cell percentage, correlation to antibody production, or delta neutralization in regards to the use of concomitant immunosuppression?

Author Response

(The authors gave the same response as above.)

Round 2

Reviewer 1 Report

All the comments raised earlier are addressed satisfactorily.